# Regulation of food intake by mechanosensory ion channels in enteric neurons

**William H Olds[1], Tian Xu[1,2,3]***

[1]Department of Genetics, Howard Hughes Medical Institute, Boyer Center for Molecular Medicine, Yale University School of Medicine, New Haven, United States; [2]National Center for International Research, Fudan-Yale Center for Biomedical Research, Fudan University, Shanghai, China; [3]Institute of Developmental Biology and Molecular Medicine, Fudan-Yale Center for Biomedical Research, Fudan University, Shanghai, China

**Abstract** Regulation of food intake is fundamental to energy homeostasis in animals. The contribution of non-nutritive and metabolic signals in regulating feeding is unclear. Here we show that enteric neurons play a major role in regulating feeding through specialized mechanosensory ion channels in *Drosophila*. Modulating activities of a specific subset of enteric neurons, the posterior enteric neurons (PENs), results in sixfold changes in food intake. Deficiency of the mechanosensory ion channel *PPK1* gene or RNAi knockdown of its expression in the PENS result in a similar increase in food intake, which can be rescued by expression of wild-type *PPK1* in the same neurons. Finally, pharmacological inhibition of the mechanosensory ion channel phenocopies the result of genetic interrogation. Together, our study provides the first molecular genetic evidence that mechanosensory ion channels in the enteric neurons are involved in regulating feeding, offering an enticing alternative to current therapeutic strategy for weight control.

***For correspondence:** tian.xu@yale.edu

**Competing interests:** The authors declare that no competing interests exist.

**Reviewing editor**: Michael Czech, University of Massachusetts Medical School, United States

## Introduction

Historically, the sensation of fullness has been documented as far back as Homer's *Odyssey*. Pioneering work by Cannon and Washburn revealed a correlation between stomach expansion and satiety in humans (*Cannon and Washburn, 1911*), which was later confirmed in rodents (*Hargrave and Kinzig, 2012*). Recently, several groups have shown that feeding-related neurons are sensitive to satiety state but not nutrients in *Drosophila* (*Marella et al., 2012*; *Dus et al., 2013*; *Pool et al., 2014*). These studies argue that non-metabolic inputs such as mechanic tension could regulate feeding.

## Results and discussion

Recent studies in *Drosophila* have identified neuronal regulation of food intake and nutrient sensing in the central nervous system (*Marella et al., 2012*; *Dus et al., 2013*). However, the potential contribution of the enteric neurons in the gastrointestinal tract has not been explored. To investigate this, we utilized four previously characterized *Gal4* lines that are expressed in enteric neurons in the different parts of the *Drosophila* digestive system (*Figure 1A,B*). While the *GMR48A05-Gal4* neurons project to the proventriculus and the anterior midgut, the *GMR51F12-Gal4* neurons project to the anterior midgut and the crop (*Figure 1A,B*) (*Jenett et al., 2012*). In contrast, both *HGN1-Gal4* and *Ilp7-Gal4* neurons project to two posterior regions in the gut: (1) the hindgut pylorus and the connecting posterior midgut and anterior hindgut, and (2) the rectum pylorus and the rectum (referred to as the posterior enteric neurons or the PENs, *Figure 1A,B*) (*Cognigni et al., 2011*).

**eLife digest** Around one third of children and two thirds of adults in the US are thought to be overweight or obese. By increasing the risk of disorders such as heart disease, stroke, diabetes and some types of cancer, obesity has become one of the leading causes of preventable death worldwide and accounts for an increasing proportion of all spending on healthcare.

Given the high costs of obesity for individuals and society, there is widespread interest in the development of drugs to aid weight loss. Three compounds are currently approved for this purpose, and they work either by reducing the body's ability to absorb fat or by acting on the brain to suppress appetite. However, all three have significant side effects.

Now, on the basis of experiments in fruit flies, Olds and Xu suggest an alternative strategy, namely targeting the 'stretch-sensitive' ion channels in the neurons in the digestive system that signal to the brain that the body has ingested enough food. By artificially activating these ion channels, it might be possible to induce feelings of fullness after smaller quantities of food have been consumed.

These ion channels—known as PPK1 ion channels—are present on posterior enteric neurons, which wrap around the muscles of the gut. Silencing these neurons caused fruit flies to eat too much, whereas activating them caused the flies to eat less. Deleting the gene that encodes the PPK1 ion channel had the same effect as silencing neurons, suggesting that drugs that act directly on PPK1 could help to regulate food intake. Consistent with this, insects ate more when their food was supplemented with a chemical that blocked the PPK1 ion channels.

By showing that PPK1 ion channels can be targeted pharmacologically, Olds and Xu have opened up a new avenue of anti-obesity research. A drug that can activate the equivalent ion channel in mammals would have the potential to aid weight loss, while avoiding the side effects associated with compounds that act directly on the brain.

We first used these *Gal4* lines to activate specific enteric neurons by expressing the temperature-sensitive ion channel, TRPA1, and measured the effects on hemolymph glucose levels (*UAS-TRPA1* [*Hamada et al., 2008*]). Activation of the *Ilp7-Gal4* neurons significantly decreases glucose levels in comparison to the *Ilp7-Gal4* and *UAS-TRPA1* controls (*Figure 1C*). Since *Ilp7-Gal4* is also expressed in the CNS and in neurons projecting to the reproductive organs (*Yang et al., 2008*), we did similar experiment using *HGN1-Gal4*, which expresses in the PENSs, but not in the other *Ilp7-Gal4* neurons (*Cognigni et al., 2011*), and obtained similar results on glucose levels (*Figure 1C*). We next examined enteric neurons projecting to other regions of the digestive system using *GMR48A05-Gal4* and *GMR51F12-Gal4* and did not observe any obvious effect (*Figure 1C*). We then assayed the effects of silencing these neurons by expressing the temperature-sensitive Dynamin, $Shi^{TS1}$ (*UAS-Shi$^{TS1}$* [*Kitamoto, 2013*]). Silencing of the PENs using both *Ilp7-Gal4* and *HGN1-Gal4* increased glucose levels in comparison to the controls, while no effect was observed by silencing the *GMR48A05-Gal4* and *GMR51F12-Gal4* neurons (*Figure 1D*). Together, the data indicate that activities of the PENs are integral in the regulation of hemolymph glucose levels.

The change of glucose levels could result from differences in food intake. Previous work by Miguel-Aliaga and colleagues revealed that silencing *Ilp7-Gal4* neurons increases defecation (*Cognigni et al., 2011*), which could be the result of increased feeding. We therefore examined the hypothesis that the activities of the PENs regulate food intake. Consistent with the hypothesis, we silenced the PENs in the absence of food and found that this abolished the gains in glucose levels (*Figure 1E*). We next investigated this directly using the capillary feeding assay (*Ja et al., 2007*). Silencing the PENs dramatically increased food intake (*Figure 2A*), which is consistent with the gains in glucose levels seen earlier. Conversely, activating these neurons caused dramatic decreases in feeding (*Figure 2B*). Together, manipulation of the activities of the PENs results in an overall six-fold change in feeding in comparison to the controls. This change is significantly larger than alterations by modulation of neuropeptide F signaling (*Hong et al., 2012*). These data indicate that the activities of the PENs play a prominent role in feeding.

As described above, experiments in mammals indicated that mechanic tension in the gastrointestinal tract could be a satiety signal (*Cannon and Washburn, 1911*; *Hargrave and Kinzig, 2012*). To

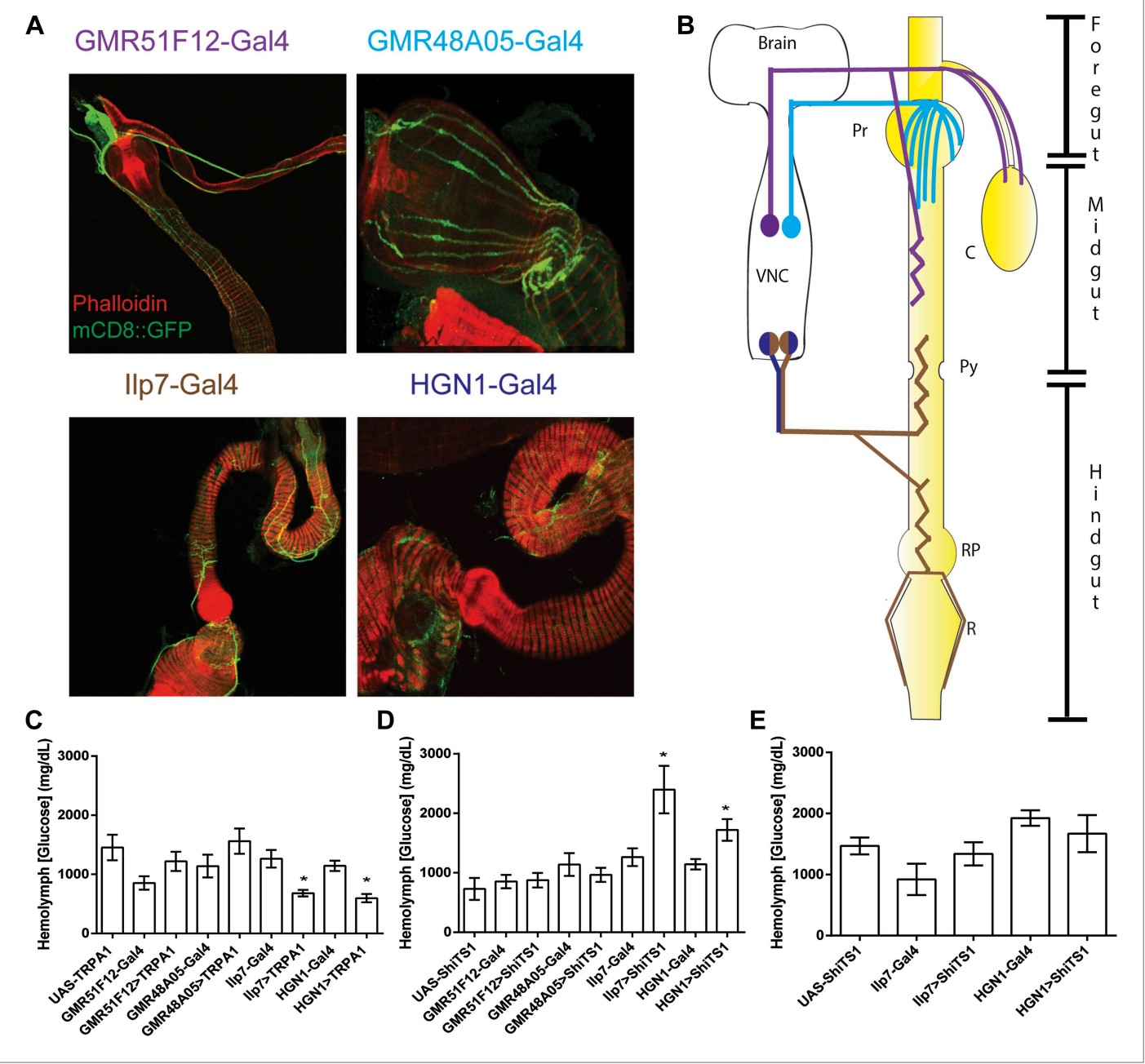

**Figure 1**. Modulating activities of Drosophila PENs causes metabolic defects. (**A**) Enteric neural projections of *Gal4* lines tested (red, phalloidin; green, *UAS-mCD8::GFP*) and their diagram (**B**). *GMR51F12-Gal4* neurons project to the foregut, anterior midgut and crop. *GMR48A05-Gal4* neurons project to the proventriculus and anterior midgut. Both *HGN1-Gal4* and *Ilp7-Gal4* drive expression in the neurons projecting to the posterior midgut, hindgut pylorus, anterior hindgut, rectal pylorus and the rectum. Pr, Proventriculus; C, Crop; Py, Pylorus; RP, Rectal Pylorus; R, Rectum; VNC, Ventral Nerve Cord. The effects of activating (**C**) or inactivating (**D**) enteric neurons on hemolymph glucose (*GMR51F12-Gal4, GMR48A05-Gal4, Ilp7-Gal4,* or *HGN1-Gal4; UAS-TRPA1 or UAS-shi*[TS1]) (*n* = 6–10 replicates of 10 flies). (**E**) The effect of silencing the PENs in starvation conditions (*Ilp7-Gal4* or *HGN1-Gal4; UAS-shi*[TS1]) (*n* = 6–9 replicates of 10 flies). * = p < 0.05, compared to corresponding *UAS* and *Gal4* control. Significances indicated are based on ANOVA and Tukey post-hoc test. Data represent the average ± s.e.m. of the results obtained.

determine whether the role of the PENs in feeding is related to mechanosensing activity, we first examined the projections of these neurons and found that they tightly wrap around the muscle layer rather than projecting into the lumen of the gut (*Figure 2C*). This anatomy favors an involvement of mechanosensory activity rather than in detecting nutritional signals in gastrointestinal tract.

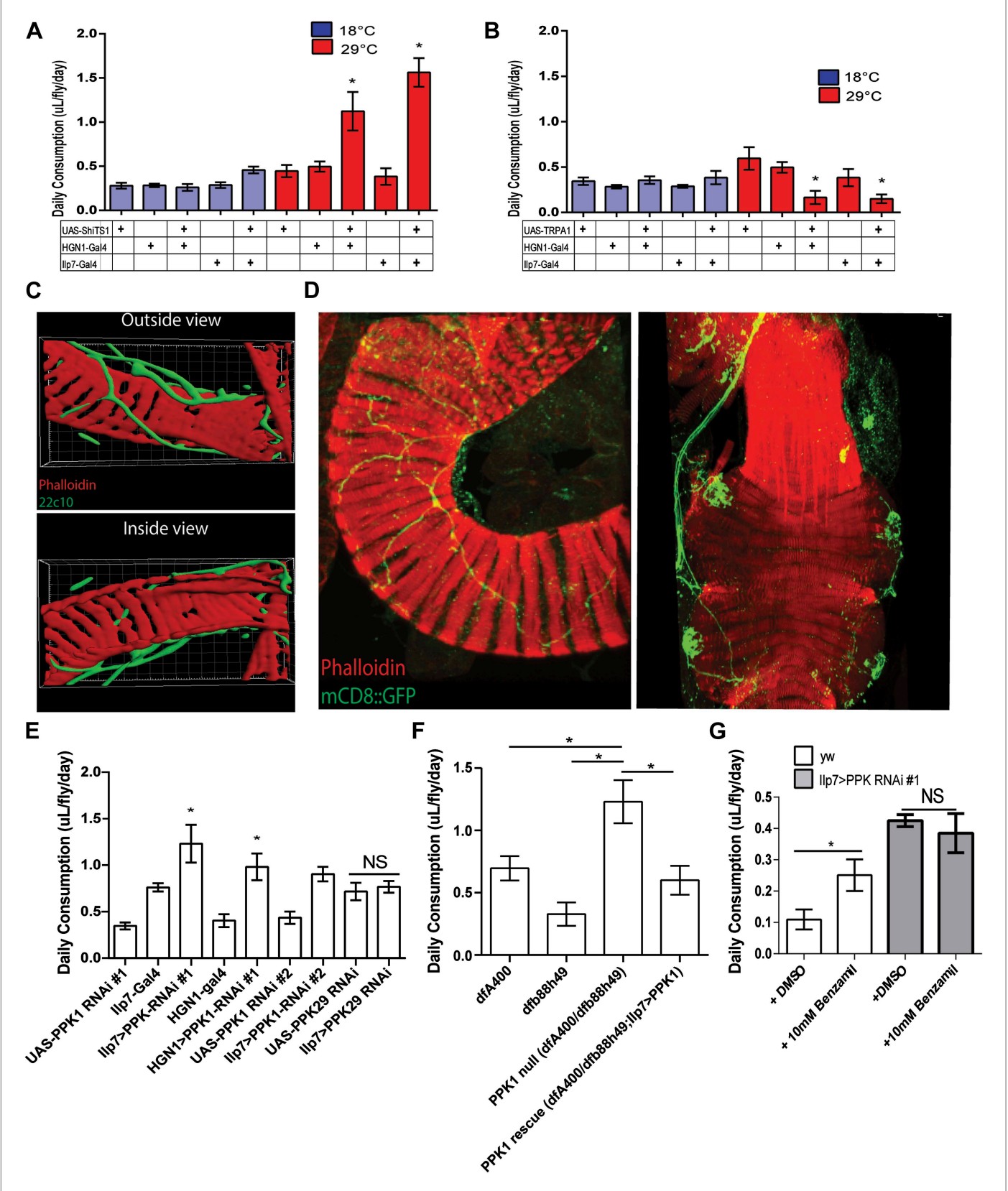

**Figure 2**. *PPK1* functions in Drosophila PENs to regulate feeding. (**A**–**B**) Results of capillary feeding assays by either inactivating (**A**, *Ilp7-Gal4* or *HGN1-Gal4; UAS-shi^{TS1}*) or activating (**B**, *Ilp7-Gal4* or *HGN1-Gal4; UAS-TRPA1*) the PENs (*n* = 4–8 replicates). (**C**) Outside and inside views of the hindgut

*Figure 2. Continued*

(red, phalloidin, muscle) with posterior enteric neuron projections (green, 22C10). (**D**) *PPK1* expresses in the PENs projecting to the hindgut pylorus (left) and rectum (right) (*PPK1-Gal4;UAS-mCD8::GFP*). (**E**) The effect of *PPK1* knock-down on food intake (*Ilp7-Gal4 or HGN1-Gal4, UAS-PPK1-RNAi#1 or UAS-PPK1-RNAi#2*) (n = 3–8 replicates). (**F**) Food intake results for *PPK1* deficiency (*dfb88h49/dfA400*) and rescued animals (*dfb88h49/dfA400; Ilp7-Gal4, UAS-PPK1*) (n = 4–7 replicates). (**G**) Food intake results when PPK1 is inhibited using benzamil in wild-type or Ilp7 > PPK1 RNAi #1 flies (n = 8–10 replicates). * = p < 0.05, compared to corresponding *UAS* and *Gal4* control or indicated controls. Significances indicated are based on ANOVA and Tukey post-hoc test. Data represent the average ± s.e.m. of the results obtained.

Previous studies in *Caenorhabditis elegans*, *Drosophila,* and mice have shown that members of the Degenerin/Epithelial Sodium Channels (DEG/ENaCs) function as a conserved family of mechanosensory ion channels (**O'Hagan et al., 2005**; **Hwang et al., 2007**; **Zhong et al., 2010**). Mutants of the *C. elegans* DEG/ENaC *Mec-4* and its *Drosophila* homolog, *PPK1*, are touch-insensitive and the affected neurons fail to generate action potentials in response to mechanic tension (**O'Hagan et al., 2005**; **Hwang et al., 2007**; **Zhong et al., 2010**). This raised the possibility that *PPK1* could be involved in regulating feeding in the PENs. We thus examined *PPK1* expression using *PPK1-Gal4* driving mCD8::GFP and confirmed its presence in the PENs (*Figure 2D*). To test whether the function of *PPK1* in the PENs is involved in the regulation of feeding, we first assayed the effect of RNAi knockdown of *PPK1* expression (*Ilp7-Gal4 or HGN1-Gal4//UAS-PPK1-RNAi*). Knockdown of *PPK1* in the PENs, but not knockdown of *PPK29*, a related family member (**Thistle et al., 2012**), dramatically increased feeding (*Figure 2E*), phenocopying the effects of silencing these neurons. Next, we examined the effect of *PPK1* deficiency on feeding and found that *PPK1* deficient flies have increased food intake (*Figure 2F*). This feeding defect is rescued by expressing a *PPK1* transgene in the PENs (*Ilp7-Gal4, UAS- PPK1* (**Ainsley et al., 2014**), *Figure 2F*). Finally, pharmacological inhibitor of DEG/ENaCs, benzamil, has been used to antagonize PPK1 in *Drosophila* and homologs in mice (**Liu et al., 2003**; **Page et al., 2007**). We therefore investigated the effect of benzamil on feeding and found it increased food consumption when supplemented in fly food, but not in flies where *PPK1* is knocked down (*Figure 2G*). Together, these data indicate that the mechanosensory ion channel, PPK1, plays a critical role in the PENs for regulating feeding.

The identification of the involvement of the DEG/ENaC mechanosensory ion channels in enteric neurons for regulating feeding lays the groundwork for investigating mechanisms underlying the phenomenon of fullness sensation. Pharmacological interventions on appetite stimulation and suppression are important for many diseases including obesity, cancer, and AIDS. Currently, all three FDA-approved weight-loss drugs have significant side-effects that target either the hypothalamus or fat absorption (**Manning et al., 2014**). Enteric DEG/ENaCs provide an attractive alternative for drug development due to their druggability, pharmacological accessibility, and fewer side-effect complications than the central nervous system. Overall, our findings indicate an important role of the DEG/ENaC mechanosensory ion channels in the enteric nervous system in food intake and suggest an exciting therapeutic alternative for fighting obesity.

## Materials and methods

### Fly stocks

Flies were reared at 25°C on standard cornmeal-molasses medium, unless indicated otherwise. The following stocks were used in this study: *GMR51F12-Gal4* (**Jenett et al., 2012**) (Bloomington Stock Center), *GMR48A05-Gal4* (**Jenett et al., 2012**) (Bloomington Stock Center), *Ilp7-Gal4* and *HGN1*-Gal4 were gifts from Dr Irene Miguel-Aliaga (**Cognigni et al., 2011**), *UAS-ShiTS1* was a gift from Toshihiro Kitamoto (**Kitamoto, 2013**), *UAS-TRPA1* (Bloomington Stock Center), 1XUAS-cd8::GFP ([**Pfeiffer et al., 2010**], Bloomington Stock Center), *PPK1-Gal4* ([**Ainsley et al., 2014**], Bloomington Stock Center), *UAS-PPK1* was a gift from Wayne Johnson (**Ainsley et al., 2008**), *UAS-PPK1 RNAi #1* (Bloomington Stock Center), *UAS-PPK1 RNAi #2* (Vienna Drosophila RNAi Center, 108683), *UAS-PPK29 RNAi* (Bloomington Stock Center), *b88h49df* (Bloomington Stock Center), *A400df* (Bloomington Stock Center) and *yw* flies.

### Hemolymph glycemia measurements

Crosses were performed at 18°C. 2-day old male flies of the indicated genotypes were incubated at 29°C in groups of ten for 24 hr and starved for 5 hr. Hemolymph was then collected as described

(*Haselton et al., 2010*) and subjected to a glucose assay (Glucose Hexokinase Liquid Stable Reagent, Thermo Scientific, Waltham, Massachusetts, USA).

## Capillary feeding assays

Flies were raised at 18°C. Capillary feeding assays were performed as described (*Ja et al., 2007*) on 2-day old males in groups of four at 29°C for 24 hr. The diet was a 5% yeast extract and 5% sucrose solution. For the benzamil experiment, male *yw* flies were provided food with 100 mM sucrose supplemented with either 10 mM benzamil or DMSO. The concentration was chosen based upon previous work (*Liu et al., 2003*). Green food coloring (1:100, McCormick, Sparks, Maryland, USA) was added to the food to track consumption.

## *Drosophila* immunohistochemistry

*Drosophila* guts were dissected and fixed as previously described (*Cognigni et al., 2011*). The following antibodies and fluorescent markers were used: rabbit Anti-GFP antibody (ab290; 1:1000; Abcam, Cambridge, UK). Alexa Fluor 488 Goat Anti-Rabbit IgG (H + L) (A11034; 1:800; Life Technologies, Gaithersburg, MD, USA), mAb22C10 (Developmental Studies Hybridoma Bank, University of Iowa), and Alexa Fluor 633 phalloidin (A22284; 1:250; Life Technologies). For the three-dimensional model of the posterior enteric neuron region, a z-stack series of confocal images were taken from a gut sample immunostained with mAb22C10 and Alexa Fluor 633 phalloidin and then converted into a model using Imaris. All images were acquired using a Zeiss LSM510 and analyzed using Imaris (Bitplane, Zurich, Switzerland).

## Statistics analyses

All data, presented as average ± s.e.m. or average ± s.d., were analyzed with GraphPad Prism 6. Unless indicated otherwise, unpaired Student's *t* test was used to determine differences between groups in each panel. For the rescue experiment, the results were compared by ANOVA followed by Tukey post hoc test. Data for each experiment met the assumption of the statistical tests. The sample size, as indicated in the figure legends, was chosen based on similar experiments reported previously, and was large enough to eliminate the variance between the groups before testing. No samples or animals were excluded from statistical analysis. All studies had been repeated for more than three times. The experimental groups were allocated randomly, and no blinding was done during allocation.

## Acknowledgements

We thank Bloomington Stock Center (NIH P40OD018537), VDRC, Wayne Johnson, Toshihiro Kitamoto, and Irene Miguel-Aliaga for fly strains, the Xu lab members for critical reading of the manuscript. WHO is a pre-doctoral fellow and supported by Genetics Training Grant T32 GM007499. TX is a Howard Hughes Medical Institute investigator.

## Additional information

### Funding

| Funder | Grant reference number | Author |
| --- | --- | --- |
| Howard Hughes Medical Institute | | Tian Xu |
| National Institutes of Health | T32 GM007499 | William H Olds |

The funders had no role in study design, data collection and interpretation, or the decision to submit the work for publication.

### Author contributions

WHO, Conception and design, Acquisition of data, Analysis and interpretation of data, Drafting or revising the article; TX, Conception and design, Analysis and interpretation of data, Drafting or revising the article

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
