## [Decision Letter]

Thank you for sending your work entitled “Regulation of Food Intake by Mechanosensory Ion Channels in Enteric Neurons” for consideration at *eLife*. Your article has been favorably evaluated by K VijayRaghavan (Senior editor), a Reviewing editor, and 2 reviewers.

The Reviewing editor and the reviewers discussed their comments before we reached this decision, and the Reviewing editor has assembled the following comments to help you prepare a revised submission.

The reviewers agree that your study makes a compelling case that in flies, afferent neurons innervating the mid to distal part of the gut play an important role in controlling feeding. In summarizing your work, one reviewer wrote: “The authors go on to show that these neurons express the mechano-sensing ion channel, PPK1, and that either neuron-selective knockdown of PPK1 or global knockout of PPK1, increase feeding. Of interest, they also show that the channel inhibitor, benzamil, increases feeding in “wild-type” flies but not in flies with neuron-specific knockdown of PPK1. These results are important and will be of significant interest to the research community.”

Thus, we are hopeful you will be able to revise your paper to overcome a major concern that was raised by both reviewers and editors. The major concern is that the mouse studies as presented are somewhat preliminary and difficult to interpret. The benzamil studies suggest that the drug modestly increases food intake and alters c-fos labeling in the brainstem. The text of the manuscript attributes this effect to alterations in “sensing” gastric distention. However, at this point in your study, it is hard to conclude this because the presumed ENaC targets of benzamil are widely expressed, including in the brain. Collectively, the reviewers felt the mouse studies may actually detract from the much more comprehensive and persuasive fly data. It is strongly recommended that either the mouse studies are eliminated from the paper or substantially clarified to overcome these concerns.

Further issues with extrapolation of these findings from flies to mice were raised related to the term “enteric”, which is confusing. In the fly studies, this term refers to neurons that have projections to the gut and send processes to the brain (as in the Figure 1–figure supplement 1). In the case of mammals, enteric neurons are entirely intrinsic to the gut. Vagal afferents, on the other hand, have projections to the gut and send sensory processes to the brain. It would seem that vagal afferents, and not the enteric nerves, would be analogous to the fly “enteric” neurons. Related to the above, it is unclear if the immune-detected channels shown in Figure 2 and in the supplementary figures are on the intrinsic enteric neurons or are on the vagal afferent neurons.

It will be very important to clarify these issues in the text and also to comment on what's similar and what's different in flies versus mice. Again, eliminating the mouse studies altogether or major clarification is required. We will be asking the reviewers to evaluate your revised manuscript, so please very carefully consider the above issues as you decide how to revise.

---

## [Author Response]

We carefully considered the issue regarding the mammalian work and decided to follow your recommendation of eliminating the mouse work from the manuscript. The rest of the manuscript for the fly work is unaltered with the exception of the changes listed below:

1) Yan Liu performed the mouse work, so she has been removed as an author.

2) We modified the Abstract to remove the mammalian work and to expand the description of the fly work.

3) Removed the description for the mammalian work in the main text (including old Figure 2 and its figure supplements, and the related methods and references).

4) In light of the recent K. Scott publication, we have cited the work and modified the sentence: “This change is significantly larger than alterations reported previously, in which the largest effect was 2.3-fold change in adult feeding by modulation of neuropeptide F signaling” to: “This change is significantly larger than alterations by modulation of neuropeptide F signaling” (see Results and Discussion sections).

5) The removal of the mammalian figures makes room for better presentation of the fly figures. This is simply achieved by combining old Figure 1 with old Figure 1–figure supplement 1 into new Figure 1, and converting the remainder of old Figure 1 into new Figure 2. This allows all the feeding-related data in Figure 2, which are separated from the non-feeding data in Figure 1.